# Neuroendocrine Biomarkers of Herbal Medicine for Major Depressive Disorder: A Systematic Review and Meta-Analysis

**DOI:** 10.3390/ph16081176

**Published:** 2023-08-18

**Authors:** Hye-Bin Seung, Hui-Ju Kwon, Chan-Young Kwon, Sang-Ho Kim

**Affiliations:** 1College of Korean Medicine, Daegu Haany University, Gyeongsan 38610, Republic of Korea; hyebin_73@naver.com (H.-B.S.); huiju313@naver.com (H.-J.K.); 2Department of Oriental Neuropsychiatry, Dong-Eui University College of Korean Medicine, Busan 47227, Republic of Korea; beanalogue@deu.ac.kr; 3Department of Neuropsychiatry of Korean Medicine, Pohang Korean Medicine Hospital, Daegu Haany University, 411 Saecheonnyeon-daero, Nam-gu, Pohang-si 790-826, Republic of Korea

**Keywords:** major depressive disorder, herbal medicine, neurobiological factor, meta-analysis, complementary integrative medicine

## Abstract

Major depressive disorder (MDD) is a medical condition involving persistent sadness and loss of interest; however, conventional treatments with antidepressants and cognitive behavioral therapy have limitations. Based on the pathogenesis of MDD, treatments using herbal medicines (HM) have been identified in animal studies. We conducted a systematic review of clinical studies to identify neurobiological outcomes and evaluate the effectiveness of HM in treating MDD. A meta-analysis was performed by searching nine databases from their inception until 12 September 2022, including 31 randomized controlled trials with 3133 participants, to examine the effects of HM on MDD using neurobiological biomarkers and a depression questionnaire scale. Quality assessment was performed using a risk of bias tool. Compared to antidepressants alone, HM combined with an antidepressant significantly increased concentrations of serotonin (SMD = 1.96, 95% CI: 1.24–2.68, *p* < 0.00001, I^2^ = 97%), brain-derived neurotrophic factor (SMD = 1.38, 95% CI: 0.92–1.83, *p* < 0.00001, I^2^ = 91%), and nerve growth factors (SMD = 2.38, 95% CI: 0.67–4.10, *p* = 0.006, I^2^ = 96%), and decreased cortisol concentrations (SMD = −3.78, 95% CI: −4.71 to −2.86, *p* < 0.00001, I^2^ = 87%). Although HM or HM with an antidepressant benefits MDD treatment through improving neuroendocrine factors, these findings should be interpreted with caution because of the low methodological quality and clinical heterogeneity of the included studies.

## 1. Introduction

Major depressive disorder (MDD) involves one or more recurrent major depressive episodes, including depressed mood, diminished interest or pleasure, weight loss, insomnia, fatigue, diminished ability to concentrate, and recurrent thoughts of death [1]. A recent systematic review of the global epidemiology of MDD showed that it has a high lifetime prevalence (2–21%) worldwide [2]. The worldwide socioeconomic burden of MDD has gradually increased, with the years lived with disabilities increasing by 36.9% between 1990 and 2010 [3]. A study published in 2019 found that MDD ranks second out of 369 diseases in terms of years lived with disability and thirteenth in disability-adjusted life years [4]. This finding is supported by the 33.7% increase in the economic burden of MDD in the United States between 2010 and 2018 [5]. In addition, MDD significantly increases the suicide risk among patients with mental and substance-use disorders and contributes the most to disability-adjusted life expectancy because of suicide [6]. Among patients with MDD, 37.7% had suicidal thoughts, 15.1% of whom attempted suicide [7]. In addition, the number of MDD cases is estimated to increase by approximately 27.6% after 2020 because of the coronavirus disease 2019 (COVID-19) pandemic [8]. The risk of MDD continues to increase, and the recurrence of COVID-19 is ongoing.

Currently, antidepressants (ADs) and cognitive behavioral therapy (CBT) are recommended as the main MDD treatments [9]. Although ADs are the most common treatment for MDD, they have limitations. The average response rate with an AD is 54%, which is not high compared to a placebo (37%) [10]. In addition, side effects have been reported, including gastrointestinal symptoms, hepatotoxicity, cardiovascular disturbances, genitourinary symptoms, central nervous system disturbances, and sleep disturbances [11]. Approximately half of the patients who discontinue or taper AD will experience withdrawal symptoms [12]. ADs are contraindicated during pregnancy and breastfeeding because it can cross the placental and blood–brain barriers and pose a risk to fetal and newborn development [13]. In addition, elderly people taking multiple drugs are at risk of adverse events associated with ADs because of multiple comorbidities and complex interactions in polypharmacy cases [14]. CBT requires considerable time and money because it is conducted through conversation and is limited by a lack of skilled practitioners able to perform the treatment [15].

Complementary and integrative medicine (CIM) for overcoming the limitations of existing treatments for mild and moderate depression has been suggested for mild and moderate depression in several clinical guidelines [16]. Clinical practice guidelines for depression include recommendations for CIM [17]. Herbal medicine (HM), a common treatment for CIM, has long been used to treat various diseases in Korea and East Asia, including China and Japan. The results show similar therapeutic effects from HM and ADs, with relatively few side effects [18].

Various hypotheses have been proposed for the pathogenesis of depressive disorders. The monoamine hypothesis, which dominates the hypotheses regarding the etiological cause of depression, states that depression is caused by the dysfunction of serotonin (5-HT) or 5-HT receptors [19]. According to the neurotrophic hypothesis, depression is caused by decreased neuroplasticity when factors related to nerve growth are damaged [20,21]. An animal model of depression revealed that HM protects nerve cells by enhancing monoamine transmission of the serotonin to activate raphe nuclei in the midbrain and inducing brain-derived growth factor expression of neurotrophins to stimulate nerve endings [22]. In addition, a randomized controlled clinical study of patients with depression found that HMs improved serotonin, brain-derived nerve growth factor, neuroendocrine factors, and depression symptoms [23,24].

Currently, a subjective questionnaire is the main method of evaluating the scale of depression. One study reported different questionnaire and biological evaluation results caused by individual differences in the interpretation of the questionnaire or errors in memory recall [25]. Therefore, a biological evaluation tool for objectively assessing depression is necessary. However, no systematic reviews on HM treatment mechanisms and biomarkers in clinical studies have been performed to date. Thus, this study systematically reviewed the use of neuroendocrine indicators in the clinical studies of HM treatment for MDD to identify objective neurobiological evaluation tools. Furthermore, in this study, a meta-analysis was conducted on the changes in neuroendocrine indicators to collate evidence regarding the effectiveness of HM treatment for MDD. The research questions investigated herein were as follows: (1) what kinds of neuroendocrine biomarkers are used in the randomized clinical trials (RCTs) employing HM for adult patients with MDD? (2) compared to antidepressants alone, is HM or the combination of HM with antidepressants effective in improving neuroendocrine indicators in adult patients with MDD?

## 2. Results

### 2.1. Study Description

Through a comprehensive search, 2702 publications were identified after 189 duplicates were removed (Figure 1). Studies were excluded by screening the title and abstract, after which 73 were included. Except for one study in which the original text could not be obtained, 72 studies underwent full-text screening. Ultimately, we considered 31 RCTs, and 31 studies [23,26,27,28,29,30,31,32,33,34,35,36,37,38,39,40,41,42,43,44,45,46,47,48,49,50,51,52,53,54,55] were excluded for the reasons listed in Figure 1.

### 2.2. Study Characteristics

#### 2.2.1. Publication Years

According to the classification by publication year, one study each was found for 2005 [52], 2007 [23], 2012 [34], and 2013 [40]. Two studies were found for 2015 [33,41], four for 2016 [26,37,48,55], three for 2017 [29,43,45], five for 2018 [30,38,44,46,47], and two for 2019 [50,51]. Seven studies were conducted in 2020 [27,28,31,32,35,42,53], two in 2021 [36,49], and two in 2022 [39,54]. All included studies were conducted in China (Appendix A).

#### 2.2.2. Study Designs

This review found five studies [23,33,34,52,53] in which HM was used alone in the treatment group and 26 studies [26,27,28,29,30,31,32,35,36,37,38,39,40,41,42,43,44,45,46,47,48,49,50,51,54,55] that combined HM and ADs. Among the studies that used HM alone, one double-blind study [23] was conducted using a placebo. Among the studies using HM and ADs in combination, two studies [37,55] were double-blinded and used a placebo. Only one 3-arm study [55] was conducted using different HM doses in the treatment group; the remaining 30 studies [23,26,27,28,29,30,31,32,33,34,35,36,37,38,39,40,41,42,43,44,45,46,47,48,49,50,51,52,53,54] were all 2-arm studies (Appendix A).

#### 2.2.3. Participants

Sixteen studies [23,26,28,30,31,33,34,38,39] suggested the Hamilton Depression Scale (HAMD) score as the study subject selection criterion, eleven studies [23,26,28,30,31,33,34] suggested only a lower limit, and five studies suggested both a lower and an upper limit. The sample size ranged from 32 to 300; the total number of participants was 3133, with an average of 101. The age of the patients was a minimum of 28.72 ± 8.79 years and a maximum of 52.7 ± 9.7 years in the treatment group and a minimum of 27.78 ± 8.56 years and a maximum of 49.16 ± 10.13 years in the control group (Appendix A).

#### 2.2.4. Interventions

Twenty-two HMs were used in the treatment interventions. The following 19 prescriptions were used: Chaifu Jieyu prescription [26], Chaihu Longgu Muli decoction [27,28], Chaihu Shugan powder [29,30,31,32], Danzhi Xiaoyao powder [23], Fuyang Shugan Juanpi prescription [33], Guipi decoction [34], Jiawei Chaihu decoction [35,36], Jiawei Xiaoyao powder [37,38,39], Jieyu Anshen decoction [40], Jieyu Anshen Dingzhi decoction [41], Jieyu pill [42], Jinkui Shenqi pill [43], Jiuwei Zhenxing granules [44], Shugan Jieyu capsules [48,49], Sini powder [50,51], Wangyou decoction [52], Xiaochaihu decoction [53], Xiaoyao pill [54], and Yueju Wan [55]. The prescription names were not listed in three cases [45,46,47]. The dosage forms included twenty-three decoctions [26,27,28,29,30,31,32,33,34,35,36,38,39,40,41,43,45,46,47,50,51,52,53], three pills [42,54,55], two powders [23,37], two capsules [48,49] and one granule [44]. Analysis of the frequency of single herbs according to HM compositions showed that *Bupleurum falcatum* Linné and *Glycyrrhiza uralensis* Fischer were the most commonly used, followed by *Paeonia lactiflora* Pallas, *Poria cocos* Wolf, and *Angelica gigas* Nakai (Table 1, Figure 2).

Eleven different types of AD were used in the control intervention. Paroxetine was used in eight studies [27,30,35,36,37,40,41,50], fluoxetine in six [33], venlafaxine in four, and escitalopram in three. Agomelatine, citalopram, doxepin, maprotiline, and sertraline were administered once; duloxetine and mirtazapine were administered twice. For the systematic classification of ADs, twenty studies used selective serotonin reuptake inhibitors (SSRIs), six used serotonin–norepinephrine reuptake inhibitors (SNRIs), three used atypical ADs, and two used tricyclic ADs (TCAs) (Appendix A).

#### 2.2.5. Treatment Periods

The treatment periods ranged from 1 to 24 weeks, averaging 7.4 weeks. No follow-ups were performed in any of the included studies.

#### 2.2.6. Outcome Measurements

The eight neuroendocrine biomarkers examined in this study were 5-HT, 5-hydroxyindoleacetic acid (5-HIAA), dopamine (DA), norepinephrine (NE), brain-derived neurotrophic factor (BDNF), neurotrophic factor (NF), nerve growth factor (NGF), and cortisol (CORT). 5-HT was used in nineteen studies [23,26,27,29,30,31,32,33,34,35,36,44,45,47,49,50,51,53,54], 5-HIAA in three [35,36,52], DA in four [45,49,53,54], NE in nine [26,31,35,36,45,47,49,53,54], BDNF in thirteen [23,28,35,36,37,38,39,40,41,42,46,48,55], NF in one [53], NGF in three [35,36,43], and CORT in six [23,27,44,47,50,51]. Four questionnaire scales were used to evaluate depression: the HAMD, the Zung Self-Rating Depression Scale (SDS), the Beck Depression Inventory (BDI), and the Montgomery–Åsberg Depression Rating Scale (MADRS). The HAMD was used in twenty-seven studies [23,26,27,28,29,31,32,33,34,35,36,37,38,39,40,41,42,43,44,45,46,47,48,50,51,52,55]; one study used only the BDI [54]. The SDS and the MADRS were used together with the HAMD.

### 2.3. Risk of Bias (RoB)

The RoB for each study is shown in Figure 3. For bias arising from the randomization process, only one study [23] was evaluated as having a low RoB; it used random number tables and randomization was performed by a third party not directly involved in the study. Four studies [34,42,52,55] wherein the randomization process had the potential to generate baseline difference were evaluated as having a high RoB. For the remaining 26 studies [26,27,28,29,30,31,32,33,35,36,38,39,40,41,43,44,45,46,47,48,49,50,51,53,54], wherein randomization was performed but the method was not mentioned or wherein group assignment was based on the order of visits but there were no baseline differences between intervention groups, were evaluated as having some concerns of RoB.

For bias due to deviations from intended intervention, all studies had a low RoB. Each participant was aware of the type of treatment they were receiving because of the characteristics of HM and AD interventions with clearly differentiated formulations. Even so, no deviation arose in the trial context.

For bias due to missing outcome data, only one study [55] was evaluated as having a high RoB, wherein dropouts were reported and the number of dropouts caused significant differences between participants of both groups.

For bias in outcome measurement, all studies had a low RoB. Even though information on blinding of the outcome assessor was lacking in all studies, it is unlikely that assessments would be influenced by their knowledge of the intervention.

For bias in the selection of the reported result, although studies did not document this protocol, 28 studies [26,27,28,29,31,32,33,35,37,38,39,40,41,43,44,45,46,47,48,50,51,54] for which all data were adequately measured and reported had some concerns over RoB. Four studies [30,36,49,53] that did not report all individual data and only reported proportions had a high RoB.

The overall RoB analysis showed a high RoB in eight studies [30,34,36,42,49,52,53,55] owing to bias from the randomization process, missing outcome data, and selection of reported results. The remaining 23 studies had some concerns over RoB.

### 2.4. Efficacy of HM Based on Neuroendocrine Biomarkers (Primary Outcome)

#### 2.4.1. HM Alone vs. ADs Alone

Four studies [23,33,34,53] using 5-HT levels of three hundred and thirty-seven participants contributed to the data. Meta-analysis of four studies comparing HM alone with AD alone indicated no significant difference in 5-HT levels (SMD = −0.05, 95% CI: −0.28 to 0.17, *p* = 0.63, I^2^ = 6%) (Figure 4).

#### 2.4.2. HM Plus ADs vs. ADs Alone

HM had significant benefits for neuroendocrine biomarkers in the meta-analysis, as assessed using 5-HT, DA, NE, BDNF, NGF, and CORT levels. The analysis of 15 studies (n = 1785) [26,27,29,30,31,32,35,36,44,45,47,49,50,51,54] showed that HM combined with ADs significantly increased concentrations of 5-HT compared to treatment with ADs alone (SMD = 1.96, 95% CI: 1.24–2.68, *p* < 0.00001, I^2^ = 97%). Subgroup analysis showed that a treatment period of 4–8 weeks had a greater effect size than the other periods. In addition, SNRIs had a greater effect size than SSRIs (Figure 5). Sensitivity analysis showed the effect size and heterogeneity decreased after six studies were excluded (SMD = 1.26, 95% CI: 0.87–1.66, *p* < 0.0001, I^2^ = 89%) (Appendix A).

The analysis of three studies (n = 606) [45,49,54] showed that HM combined with ADs significantly increased concentrations of DA compared to treatment with ADs alone (SMD = 1.49, 95% CI: 1.12–1.85, *p* < 0.00001, I^2^ = 73%) (Figure 5). The analysis of eight studies (n = 1126) [26,31,35,36,45,47,49,54] showed that HM combined with ADs significantly increased concentrations of NE compared to ADs alone (SMD = 1.24, 95% CI: 0.80–1.69, *p* < 0.00001, I^2^ = 91%). Subgroup analysis showed that SSRIs and SNRIs had greater effect sizes than atypical ADs (Figure 5).

The analysis of 12 studies (n = 1066) [28,35,36,37,38,39,40,41,42,46,48,55] showed that HM combined with AD significantly increased concentrations of BDNF compared to ADs alone (SMD = 1.38, 95% CI: 0.92–1.83, *p* < 0.00001, I^2^ = 91%) (Figure 6). Sensitivity analysis showed the effect size and heterogeneity decreased slightly after three studies were excluded (SMD = 1.11, 95% CI: 0.84–1.39, *p* < 0.0001, I^2^ = 84%) (Appendix A). The analysis of three studies (n = 248) [35,36,43] showed that HM combined with ADs significantly increased the concentrations of NGF compared to ADs alone (SMD = 2.38, 95% CI: 0.67–4.10, *p* = 0.006, I^2^ = 96%) (Figure 7). The analysis of five studies (n = 462) [27,44,47,50,51] showed that HM combined with ADs significantly decreased CORT concentrations compared to ADs alone (SMD = −3.78, 95% CI: −4.71 to −2.86, *p* < 0.00001, I^2^ = 87%). Subgroup analysis showed that SSRIs had a greater effect size than the other ADs (Figure 7). Sensitivity analysis showed the effect size and heterogeneity decreased after one study was excluded (SMD = −3.08, 95% CI: −3.48 to −2.67, *p* < 0.00001, I^2^ = 52%) (Appendix A).

### 2.5. Efficacy of HM Based on Questionnaire Evaluation Scales (Secondary Outcome)

#### 2.5.1. HM Alone vs. ADs Alone

Four studies [23,33,34,53] using the HAMD with two hundred and sixty-six participants contributed to the data analysis. Meta-analysis of four studies comparing HM alone with AD alone showed no significant difference in the HAMD score (SMD = −0.44, 95% CI: −1.28 to 0.40, *p* = 0.30, I^2^ = 91%) (Appendix A).

#### 2.5.2. HM plus ADs vs. ADs Alone

The meta-analysis of 22 studies (n = 2121) [27,28,29,31,32,35,37,38,39,40,41,42,43,44,45,46,47,48,50,51,52,55] found that HM with ADs significantly reduced the HAMD score compared with ADs alone (SMD = −1.81, 95% CI: −2.23 to –1.38, *p* < 0.00001, I^2^ = 94%). Subgroup analysis found no significant differences between the two groups when the treatment period was <4 weeks. SNRIs and atypical ADs showed effect sizes similar to SSRIs (Appendix A). Sensitivity analysis showed the effect size and heterogeneity decreased after 11 studies were excluded (SMD = −1.45, 95% CI: −1.75 to −1.14, *p* < 0.00001, I^2^ = 81%) (Appendix A).

### 2.6. Safety of HM

Of the twenty-one studies [27,28,30,31,33,35,36,37,39,41,42,43,44,45,46,47,49,50,51,52,53] reporting adverse reactions, two [39,42] used the Treatment Emergent Symptom Scale, fifteen [27,28,30,31,33,36,37,43,44,45,47,49,50,51,53] specified the number of patients with adverse events, and four [35,41,46,52] did not specify adverse events or the number of patients involved. The analysis of the two studies (n = 194) [34,53] of HM alone found no significant difference in the occurrence of adverse events between using HM alone and ADs alone (RR = 0.19, 95% CI: 0.01–3.15, *p* = 0.25, I^2^ = 89%) (Appendix A). The analysis of 13 studies (n = 1522) [27,28,30,31,36,37,43,44,45,47,49,50,51] using HM with ADs showed that HM plus ADs significantly improved the incidence of adverse events compared to the ADs alone (RR = 0.48, 95% CI: 0.38–0.62, *p* < 0.00001, I^2^ = 1%) (Appendix A).

### 2.7. Publication Bias

A funnel plot was calculated based on the 5-HT, BDNF, and HAMD of RCTs using HM and ADs in combination (Figure 8). For 5-HT, the funnel plot was asymmetric, suggesting a possibility of publication bias, with a value of *p* = 0.019 in Egger’s test. For BDNF, the funnel plot was symmetric, but a possibility of publication bias was suggested, with a value of *p* = 0.005 in Egger’s test. In the HAMD, the funnel plot was asymmetric, but no possibility of significant publication bias was indicated, with a value of *p* = 0.685 in Egger’s test.

## 3. Discussion

### 3.1. Summary of Evidence

In this study, 31 RCTs using neuroendocrine biomarkers were included through a comprehensive search among clinical studies of HM treatment for MDD. The neuroendocrine biomarkers used were neurotransmitters (5-HT, 5-HIAA, DA, and NE), neurotrophic factors (BDNF, NF, and NGF), and hypothalamic-pituitary-adrenal hormone (CORT); among these, 5-HT and BDNF are the main indicators. Five studies were identified [23,33,34,52,53] in which HM was used alone in the treatment group; twenty-six studies [26,27,28,29,30,31,32,35,36,37,38,39,40,41,42,43,44,45,46,47,48,49,50,51,54,55] used HM and ADs in combination. Various prescriptions were used in the included studies; however, prescriptions involving *Bupleurum falcatum* Linné were predominant. In the HM prescriptions included, *Bupleurum falcatum* Linné, *Glycyrrhiza uralensis* Fischer, *Paeonia lactiflora* Pallas, *Poria cocos* Wolf, and *Angelica gigas* Nakai were frequently used. SSRIs were the most common ADs used as control interventions, with paroxetine being the most frequently used. The treatment duration was usually 4–8 weeks. The meta-analysis showed that HM alone improved 5-HT and HAMD scores, similar to ADs. Both neuroendocrine biomarkers (5-HT, DA, NE, BDNF, NGF, and CORT) and HAMD scores were significantly improved in the HM plus ADs group compared to the ADs-alone group. In addition, the incidence of adverse reactions in the HM plus ADs group was significantly lower than that of ADs alone. The performance RoB of the included studies was high because most studies did not use a placebo; the methodological quality of the overall study was low because the bias risk for selection, detection, and selective reporting were evaluated as uncertain. Publication biases were identified for 5-HT and BDNF in the HM combined with ADs group.

### 3.2. Neuroendocrine Mechanisms of HM in the Treatment of MDD (Table 2)

#### 3.2.1. Monoamine Neurotransmitters

5-HT is a major neurotransmitter in the brain, and patients with depression have significantly lower platelet serotonin concentrations than healthy individuals [56]. Serotonin receptor dysfunction is the main cause of MDD; 5-HT membrane transporter protein regulates 5-HT production and is a major target of SSRIs [57]. The meta-analysis of 5-HT in this study showed that HM alone increased 5-HT concentrations and improved HAMD scores, similar to ADs alone. In addition, HM combined with ADs significantly increased 5-HT concentrations and improved HAMD scores compared with ADs alone. This finding suggests that HM used for MDD can exhibit AD effects by acting on the 5-HT pathway and receptors, similar to SSRIs. The Zhike–Houpu herbal pair showed AD effects in behavioral experiments on depression-model rats, and the mechanism was related to hippocampal serotonin receptor 1A, which regulates the release of other neurotransmitters [58]. In addition, Xiaoyao powder administration significantly increased the concentrations of 5-HT in the blood of depression-model rats but showed no significant change in normal rats. This result suggests that Xiaoyao powder acts selectively on abnormal 5-HT receptors [59]. 5-HIAA is the first metabolite of 5-HT, and the 5-HT/5-HIAA ratio is used to indicate 5-HT concentration reversal [60]. Two included studies reported decreased plasma concentrations of 5-HIAA and increased plasma concentrations of 5-HT [35,36]. Danzhi Xiaoyao powder increased the concentration of 5-HT in the hippocampus of depression-model rats and improved the 5-HT/5-HIAA ratio [61].

DA is a neurotransmitter involved in motor control, cognitive behavior, and emotion and is closely related to anhedonia or amotivation among depression symptoms [62]. Rats exposed to chronic stress in animal models of depression show reduced DA activity in the limbic region compared with normal rats [63]. In this study, HM and AD combination treatment significantly increased DA concentrations and significantly improved HAMD scores compared to AD treatment alone. This finding is consistent with the finding that white pine increases the concentrations of DA and NE in depression-model rats and effectively improves their depressive behavior [64]. NE is also a major neurotransmitter in the brain, and imipramine, the first TCA, exerts AD effects by enhancing the synaptic activity of NE [65]. In this study, HM plus AD treatment significantly improved HAMD scores and increased NE concentrations compared to AD treatment alone. This finding is consistent with the finding that the Ganmai Dazao decoction increased the concentration of NE in menopausal patients with depression, similar to ADs [61]. In other words, HM exhibits AD effects by improving 5-HT–related metabolism and regulating the activities of DA and NE.

#### 3.2.2. Neurotrophic Factors

BDNF is a neurotrophin secreted from both peripheral and central nerves by target cells or astrocytes. Decreases in plasma and serum BDNF levels have been observed in patients with depression [64]. BDNF has been implicated in psychiatric disorders, including depression, because of its important role in brain development and neuroplasticity [65]. BDNF binds to tropomyosin receptor kinase B and activates the cAMP response element-binding protein (CREB) via three major pathways to propagate AD effects [65]. The meta-analysis of BDNF in this study showed that treatment with HM combined with ADs significantly increased BDNF concentrations and improved HAMD scores compared with treatment with ADs alone. Chaihu Shugan San and Gan-Mai-Da-Zao decoctions have shown AD effects by improving the BDNF–CREB signaling system in the hippocampus and prefrontal cortex in a rat model [66,67].

NGF is also a neurotrophin, and serum NGF levels are low in patients with depression [68]. NGF exerts AD effects by increasing the concentrations of extracellular signal–regulated kinase (ERK) and CREB in the hippocampus [69]. HM combined with ADs significantly increased NGF concentrations and improved HAMD scores compared to ADs alone. The herbal mixture of *Sesami Semen Nigrum* and *Longan Arillus* exhibits AD effects in rat models of depression through a mechanism related to the NGF-induced signal transduction system [70]. Ginger also shows AD effects by improving the NGF-ERK-CREB signaling system in rat models of depression [71]. Thus, HM can act on various pathways to increase BDNF and NGF levels and exhibit AD effects.

#### 3.2.3. Hypothalamic–Pituitary–Adrenal (HPA) Axis Hormones

CORT is a hormone released from the adrenal cortex that constitutes the HPA axis and shows a circadian pattern of low concentrations in the morning and high concentrations in the afternoon [72]. Patients with MDD show greater afternoon CORT concentrations than non-depressed individuals [73]. In addition, corticotropin-releasing factor (CRF) secreted from the hypothalamus is associated with the NE system, resulting in the enhancement of the HPA axis and a decrease in NE when chronic stress is stimulated [74]. HM combined with ADs significantly decreased CORT concentrations and improved HAMD scores compared to ADs alone. Danzhi Xiaoyao powder and Kai Xin San inhibit the hyperactivity of the HPA axis and reverse abnormal activity in depression-model rats, significantly reducing the concentrations of CORT, ACTH, and corticotropin-releasing factor hormone (CRH) in the plasma and hypothalamus [61,75]. Thus, HM can exert AD effects by improving the stress-related endocrine factors of the HPA axis.

### 3.3. Clinical Implications

This study confirmed the neuroendocrine biomarkers used in clinical studies of HM for depression. Therefore, the results of this study can be used as basic data for objectively evaluating the treatment effects in patients with depression treated with HM in clinical practice [76]. Currently, subjective questionnaire scales are the primary method for evaluating the severity of depression. Objective biomarkers can help determine the severity of depression more accurately by supplementing questionnaire evaluations. In addition, our findings can be used in future clinical studies along with neuroendocrine biomarkers. Conducting such studies will provide more-objective evidence on the AD effects of HMs and help in decision-making related to their use in clinical settings.

### 3.4. Limitations and Implications for Further Research

This review has several limitations. First, almost all of the included studies had low methodological quality. Except for three studies, double blinding was not conducted, and no information on allocation concealment was presented. Therefore, the results of this study are at a high risk of bias, and the effect estimate may be exaggerated. Second, in the publication bias analysis, asymmetry was observed in the 5-HT and BDNF levels; therefore, care should be taken when interpreting the results. Third, all the included studies were conducted in China; therefore, our results may be difficult to generalize to patients from other countries. In addition, the placebo effect according to the characteristics of the Chinese medical system and cultural preferences for Oriental medicine should be considered [77]. Fourth, sufficient follow-up was not conducted to determine the treatment continuation. Fifth, the study did not cover the pathogenesis of all depressive disorders. Only neuroendocrine indicators that can be confirmed through blood tests were targeted; therefore, indicators of the neuroimmune system and metabolic alteration were not included. In addition, because no studies on CRF and ACTH have been conducted, explanations of the HPA axis stabilization mechanism using only the study results on CORT are insufficient. Therefore, studying the pathogenesis of depressive disorders and exploring biomarkers is necessary to evaluate improvements in depressive disorders from HM treatment. Sixth, whether the improvement in neuroendocrine biomarkers by HM combinations in this study happened because of an increase in the concentration of ADs, because of drug interactions with HM, or because of HM’s multiple ingredients acting on multiple targets through various pathways is unclear [78].

### 3.5. Implications for Further Research

The suggestions for future research are as follows. First, large-scale multicenter RCTs with high methodological quality should be conducted to confirm the results of this study. Designing RCTs in countries other than China is also necessary, including Korea, Japan, and the United States of America. Second, studies analyzing the mechanism of HM treatment should be conducted using biomarkers covering the pathogenesis of depressive disorders, possibly suggesting how HM induces therapeutic effects in patients with MDD. Furthermore, if large amounts of high-quality research data are accumulated, presenting clinical practice guidelines with greater evidence will be possible. Third, confirming that the AD effects of HM continue is necessary in future studies. A long-term follow-up study should be conducted to determine the duration of the AD effects of HM.

## 4. Materials and Methods

### 4.1. Study Registration

The study protocol is registered in PROSPERO (registration number: CRD42022358944). This review followed the 2020 Preferred Reporting Items for Systematic Reviews and Meta-Analyses statement [79].

### 4.2. Inclusion and Exclusion Criteria

#### 4.2.1. Types of Included Studies

The study included only RCTs that used HMs to treat depression. Case reports, cross-sectional studies, pilot studies, feasibility studies, simple reviews, mechanisms, and experimental studies were excluded from the analysis. The study had no publication or language restrictions, but all search terms were written in Korean, English, Chinese, or Japanese.

#### 4.2.2. Participants

The study included patients aged >18 years diagnosed with MDD and analyzed only studies that used specific diagnostic criteria for depression, including the Diagnostic and Statistical Manual of Mental Disorders, the International Classification of Diseases, and the Chinese Classification of Mental Disorders. Patients in studies that did not provide diagnostic criteria were excluded, as were patients with other diseases, including heart disease, stroke, cancer, schizophrenia, and bipolar disorder. The study had no restrictions on age, sex, nationality, or race.

#### 4.2.3. Types of Intervention

Studies using HM alone or HM combined with ADs and studies using HM and a placebo were examined. Only studies for which the composition or ingredients of the HM were confirmed were included. Cases using a single herb instead of a complex HM were excluded, as were studies using other CIMs, such as acupuncture/acupressure or moxibustion. The forms or volumes of the HM had no restrictions. Studies that used only an AD or placebo as a control intervention were also included.

#### 4.2.4. Outcome Measures

Since various neurobiological indicators are used for the evaluation of depression, this study focused only on neuroendocrine biomarkers used in clinical studies of HM for depression. The primary outcomes were the neuroendocrine biomarkers, including the 5-HT, 5-HIAA, DA, NE, BDNF, NF, NGF, CORT, adrenocorticotropic hormone, and corticotropin-releasing factor. Secondary outcomes were depression questionnaire evaluation scales, including the HAMD, SDS, BDI, and MADRS.

### 4.3. Search Methods

Nine electronic bibliographic databases were searched from their inception dates to 12 September 2022 by two independent reviewers (H.-B.S. and H.-J.K.), including PubMed, Embase via Elsevier, the Cochrane Central Register of Controlled Trials, the Allied and Complementary Medicine Database via EBSCO, PsycARTICLES via ProQuest, the Oriental Medicine Advanced Searching Integrated System, the Korea Citation Index, the China National Knowledge Infrastructure, and Citation Information by NII. Google Scholar was used to conduct gray literature searches. References from relevant systematic reviews and the retrieved articles were searched manually. The authors of the published papers were contacted when the electronic files could not be accessed. The search terms and strategies are detailed in Appendix A.

### 4.4. Data Collection and Quality Assessment

#### 4.4.1. Literature Selection

All studies retrieved using the search strategy were imported into EndNote 20.2.1 (Clarivate Analytics; Boston, MA, USA); duplicate studies were eliminated. The titles and/or abstracts of the retrieved studies and those from additional sources were screened independently by two researchers (H.-B.S. and H.-J.K.) to identify studies that potentially met the inclusion criteria. Subsequently, the full texts of the potentially eligible studies were retrieved and assessed by the same two independent researchers. If a discrepancy occurred, disagreements regarding study eligibility were resolved through discussion with a third experienced review author (S.-H.K.).

#### 4.4.2. Data Extraction

Two authors (H.-B.S. and H.-J.K.) independently extracted data from the selected studies. Disagreements were resolved by discussion or consulting an experienced review author (S.-H.K.) if no consensus was achieved. We used a standardized data extraction form that included the source, author, publication year, study design, participant characteristics, intervention, comparator, duration, follow-up, outcome measurements, results, and adverse events. Microsoft Excel version 1808 (Microsoft Corp., Redmond, WA, USA) was used for data and information management. Missing data, incomplete information, or data errors were collected, including data on withdrawals and exclusions, and the corresponding author was asked via email or telephone for the correct information.

#### 4.4.3. Assessment of the RoB and Quality of Included Studies

The provided Cochrane RoB tool for randomized trials (RoB 2.0) [80] was used to analyze the RoB in the selected studies. Two researchers (H.-B.S. and H.-J.K.) independently evaluated the study quality. The following factors were considered: randomization process, deviations from the intended interventions, missing outcome data, outcome measurement, and selection of the reported result. The questions were answered as follows: yes, probably yes, probably no, no, and no information. The RoB was graded as high, low, or some concerns. When the two reviewers could not reach a consensus, the issue was resolved by consensus with an experienced review author (S.-H.K.).

### 4.5. Statistical Analysis

#### 4.5.1. Strategy for Data Synthesis

A descriptive analysis of the included studies was performed, including participant demographics, specifics of the experimental and control interventions, outcomes, results, and adverse events. The results were pooled using RevMan version 5.4 (Cochrane; London, UK) to calculate the mean difference (MD) if the same types of intervention, comparison, and outcome measurement were used, the standard mean difference (SMD) for continuous outcomes, the risk ratio (RR) for binary outcomes, and 95% confidence intervals (CI). The true effect size was assumed to vary from study to study. The studies were assumed to be a random sample of observable effect sizes because the clinical characteristics of the patients with depression across the studies were significantly heterogeneous. The data were pooled using a random effects model regardless of heterogeneity, based on the I^2^ statistic. However, the data were pooled using a fixed-effects model if very few studies were included in the meta-analysis, indicating that the estimate of the between-study variance lacked precision.

#### 4.5.2. Subgroup Analysis

A subgroup analysis was performed according to the treatment duration. Because the effect on neuroendocrine biomarkers differed depending on the intervention, a subgroup analysis was conducted according to the AD class.

#### 4.5.3. Publication Bias and Sensitivity Analysis

A funnel plot was constructed if more than 10 studies were included, and symmetry was evaluated. Otherwise, Egger’s test was performed to reduce visual errors. Funnel plotting and Egger’s test were performed using Stata 16.0 with metan cord and metafunnel. The methodology was the same as that reported by Shim et al. [81]. In addition, a sensitivity analysis was performed by excluding studies that were rated as having a high RoB or that were numerically distant from the rest of the data.

## 5. Conclusions

Based on the systematic literature review and meta-analysis of HM treatment for MDD using neuroendocrine biomarkers, the following conclusions were drawn:HM alone showed improvements similar to ADs for neuroendocrine biomarkers (5-HT) and the depression questionnaire scale (HAMD).HM combined with ADs significantly improved neuroendocrine biomarkers (5-HT, BDNF, DA, NE, NGF, and CORT) and the depression questionnaire scale (HAMD) compared with ADs alone.HM combined with ADs had a significantly lower incidence of adverse events than ADs alone.HM can treat depression by improving the expression of a patient’s neurotransmitters, neurotrophic factors, and HPA-axis hormones.In the future, conducting a high-quality multicenter large-scale RCT study using various neuroendocrine biomarkers is necessary.

## Figures and Tables

**Figure 1 pharmaceuticals-16-01176-f001:**
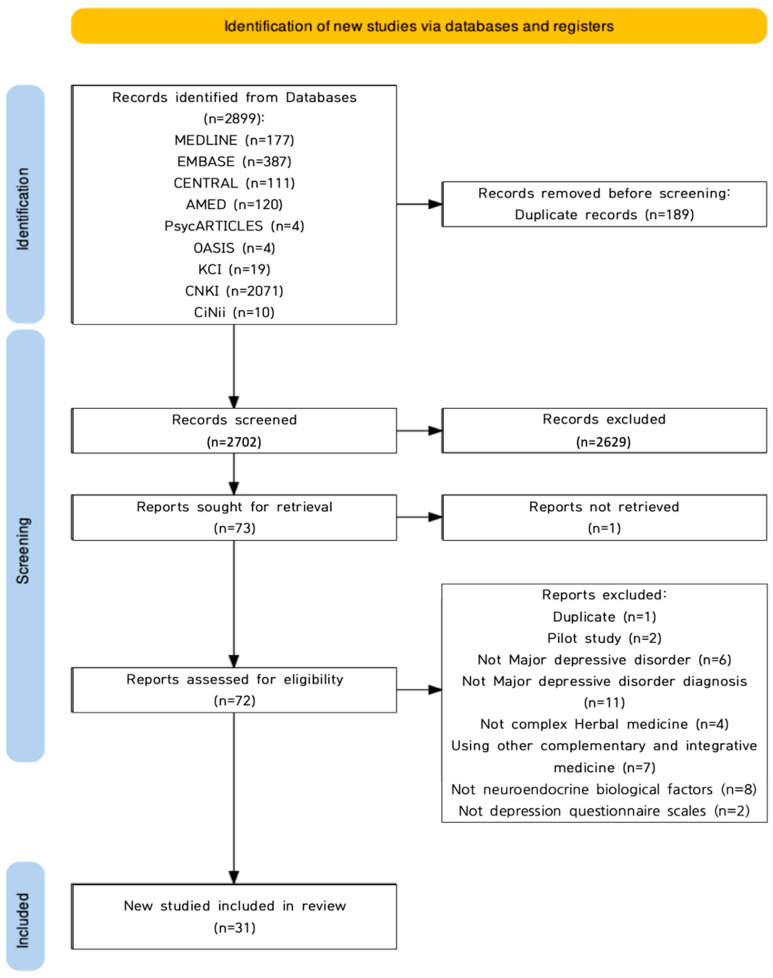
Flowchart of identification and screening for the eligible studies.

**Figure 2 pharmaceuticals-16-01176-f002:**
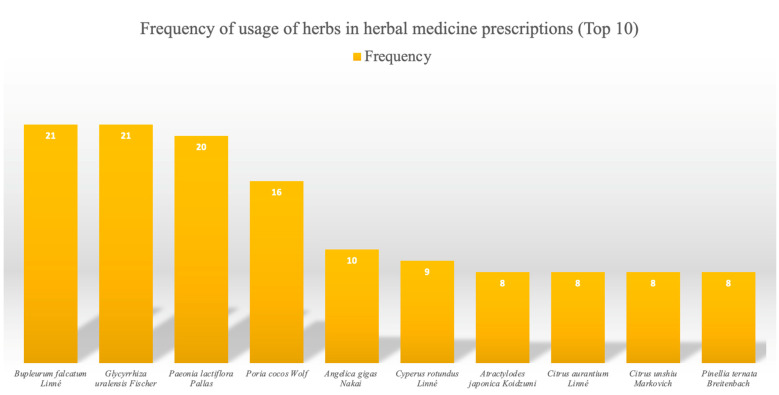
Frequency of herbs used in herbal medicine prescriptions (Top 10).

**Figure 3 pharmaceuticals-16-01176-f003:**
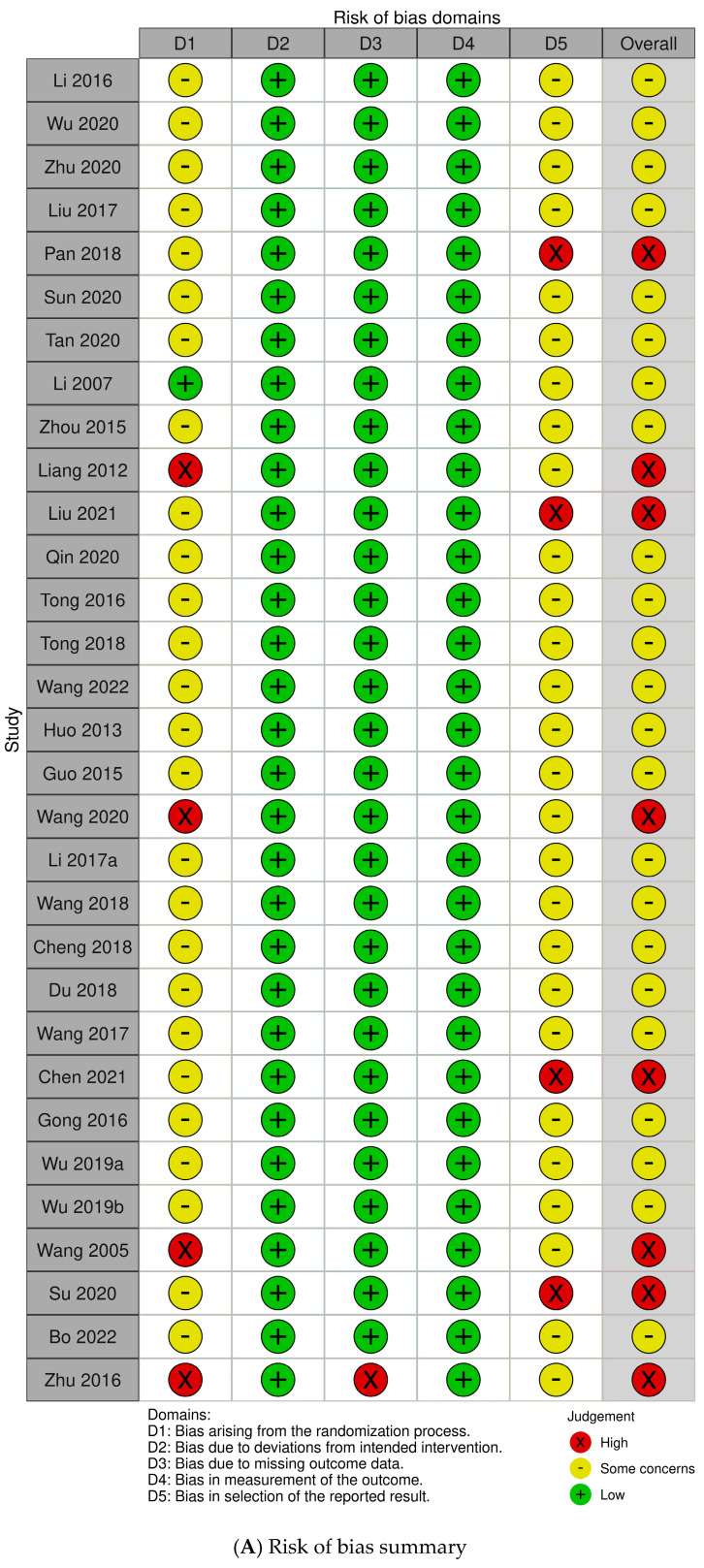
(**A**) Risk of bias summary. Low, unclear, and high risk, respectively, are represented using the following symbols: “+”, “?”, and “−”. (**B**) Risk of bias graph. Review of authors’ judgments about each risk-of-bias item presented as percentages across all included studies.

**Figure 4 pharmaceuticals-16-01176-f004:**
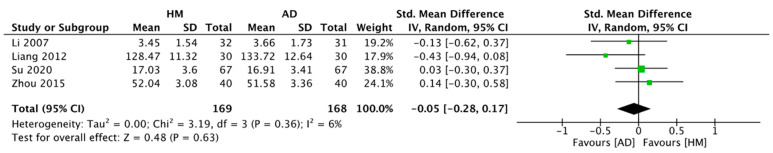
Forest plot of the comparison between herbal medicine versus antidepressants assessing 5-HT. AD, antidepressant; HM, herbal medicine.

**Figure 5 pharmaceuticals-16-01176-f005:**
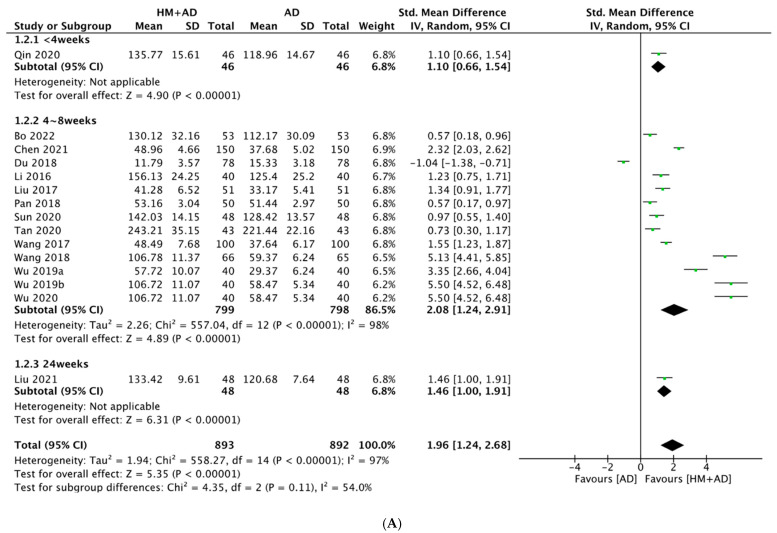
Forest plot of the comparison between herbal medicine plus antidepressants versus antidepressants alone assessing (**A**) 5-HT, subgroup analysis according to the duration of treatment; (**B**) 5-HT, subgroup analysis according to the class of AD; (**C**) DA; (**D**) NE, subgroup analysis according to the duration of treatment; (**E**) NE, subgroup analysis according to the class of AD. AD, antidepressant; DA, dopamine; HM, herbal medicine; NE, norepinephrine.

**Figure 6 pharmaceuticals-16-01176-f006:**
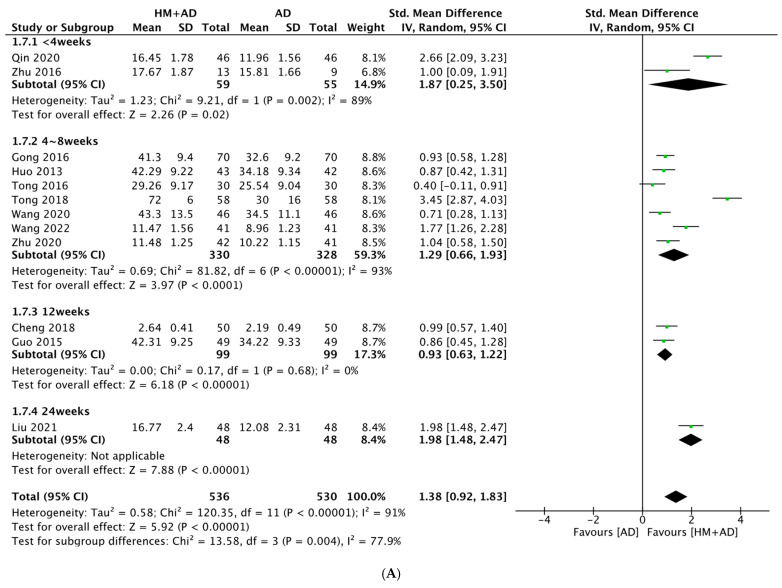
Forest plot of the comparison between herbal medicine plus antidepressants versus antidepressants alone assessing (**A**) BDNF, subgroup analysis according to the duration of treatment; (**B**) BDNF, subgroup analysis according to the class of AD; (**C**) NGF, subgroup analysis according to the duration of treatment. AD, antidepressant; BDNF, brain-derived neurotrophic factor; HM, herbal medicine; NGF, nerve growth factor.

**Figure 7 pharmaceuticals-16-01176-f007:**
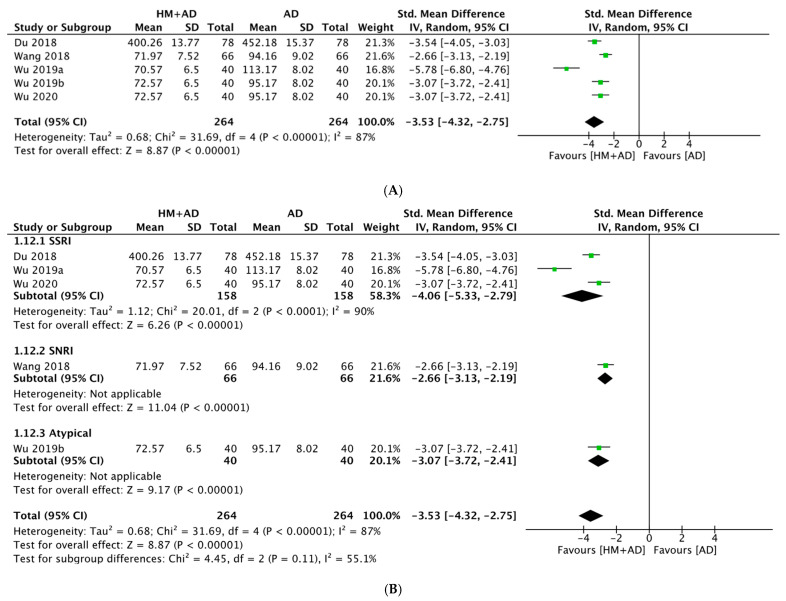
Forest plot of the comparison between herbal medicine plus antidepressants versus antidepressants alone assessing (**A**) CORT; (**B**) CORT, subgroup analysis according to the class of AD. AD, antidepressant; CORT, Cortisol; HM, herbal medicine.

**Figure 8 pharmaceuticals-16-01176-f008:**
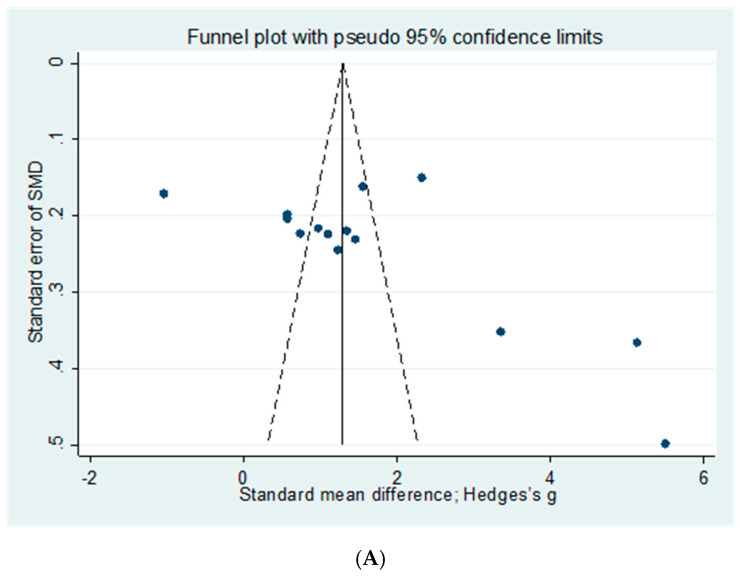
Funnel plot of the comparison of herbal medicine plus antidepressants versus antidepressants alone assessing (**A**) 5-HT, (**B**) BDNF, (**C**) HAMD. 5-HT, Serotonin; BDNF, Brain-derived neurotrophic factor; HAMD, Hamilton Depression Scale.

**Table 1 pharmaceuticals-16-01176-t001:** Frequency of usage of herbs in herbal medicine prescriptions (more than once).

Frequency	Herb
21	*Bupleurum falcatum* Linné, *Glycyrrhiza uralensis* Fischer
20	*Paeonia lactiflora* Pallas
16	*Poria cocos* Wolf
10	*Angelica gigas* Nakai
9	*Cyperus rotundus* Linné
8	*Atractylodes japonica* Koidzumi, *Citrus aurantium* Linné, *Citrus unshiu* Markovich, *Pinellia ternata* Breitenbach
7	*Curcuma aromatica* Salisb, *Polygala tenuifolia* Willdenow, *Zizyphus jujuba* Mill
6	*Cnidium officinale* Makino
5	*Scutellaria baicalensis* Georgi
4	*Citrus aurantium* Linné, *Gardenia jasminoides* Ellis, *fossilia ossis mastodi*
3	*Albizia julibrissin* Durazz, *Aucklandia lappa* Decne, *Polygonum multiflorum* Thumb, *Cinnamomum cassia* Presl, *Ostrea gigas* Thunberg, *Lilium lancifolium* Thunb, *Hypericum perforatum, Paeonia suffruticosa* Andrews, *Panax ginseng* C. A. Meyer, *Rheum palmatum* Linne, *Zizyphus jujuba* Miller var. *inermis Rehder*
2	*Aconitum carmichaeli* Debeaux, *Acorus gramineus* Solander, *Astragalus membranaceus* Bunge, *Platycladus orientalis* Franco, *Codonopsis pilosula* Nannfeldt, *Mentha arvensis* Linné var. *piperascens* Malinvaud ex Holmes, *Salvia miltiorrhiza* Bunge, *Zingiber officinale* Roscoe

**Table 2 pharmaceuticals-16-01176-t002:** Overview of the herbal medicine treatment for MDD.

Main Mechanism	Outcome Raised	Outcome Reduced
Monoamine neurotransmitter	5-HT, DA, NE	5-HIAA
Neurotrophic factor	BDNF, NGF, NF	-
HPA-axis hormone	-	CORT

Note: 5-HIAA = 5-hydroxyindoleacetic acid; 5-HT = 5-hydroxytryptamine (serotonin); BDNF, brain-derived neurotrophic factor; CORT, cortisol; DA, dopamine; NE, norepinephrine; NGF, nerve growth factor.

## Data Availability

The search strategy presented in the methodology section and on the flow chart. Any further inquiries could be referred to the corresponding author.

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
