# Peer review of "Neuroendocrine Biomarkers of Herbal Medicine for Major Depressive Disorder: A Systematic Review and Meta-Analysis"

_pharmaceuticals, 2023, doi:10.3390/ph16081176_

Round 1

Reviewer 1 Report

The study aimed to summarise the existing literature and prepare a systematic review with meta-analysis on herbal medicine treatment for Major depression based solely on RCT studies.

Overall it is a very interesting topic, and the manuscript is, in many areas, well done. However, after going deeper into the content, more issues occur. I will provide a comprehensive list of the main problems that I was able to detect:

1.      Please use the risk of bias assessment tool 2.0 (the new version is standard after 2016 – please use Excel tool for assessment (https://www.riskofbias.info/welcome/rob-2-0-tool/current-version-of-rob-2 ) and robvis for visualisation (https://www.riskofbias.info/welcome/robvis-visualization-tool ))

2.      Because this is a systematic review, the most important step is to check the method section before reading the rest of the data. Because of that, please provide the full name of the abbreviation used in the method section or, alternatively, provide the list of used abbreviations at the beginning of the manuscript.

3.      Please formulate the research question and provide it with the overall aim of the study at the end of the introduction.

4.      Because the authors present a lot of data, especially forest plots, I believe Table 1 could be moved into the supplementary file.

5.      Table 1 – please define A and B in the Note below the table. Providing it only in the column description is insufficient. Moreover, because this table is huge, the first row should also be displayed on every page.

6.      Please provide content from Table 2 in the form of some figure/graph for better display.

7.      2.3 RoB section needs to be rewritten to meet RoB 2.0 standard

8.      Figure 2 – needs to be changed for RoB 2.0

9.      Figure 3 – because I2 is below 50% in this situation, a fixed effect model with the Mantel-Haenszel method should be used. Other statistics are correct regarding using a Random model with inverse variance.

10.  All of the RevMan results – in The RevMan you could change the name of the columns in the table section to be the same as in the plot section, eg. now you have Experimental/Control and Favours AD/Favours HM+AD – it should be the same.

11.  Figures 6 and 7 – please use other software or omit this section completely because in RevMan when the random model is used, the funnel plot does not properly display the plot area. We are not able to see any funnel. 

Author Response

Comment #1:

Please use the risk of bias assessment tool 2.0 (the new version is standard after 2016 – please use Excel tool for assessment (https://www.riskofbias.info/welcome/rob-2-0-tool/current-version-of-rob-2 ) and robvis for visualisation (https://www.riskofbias.info/welcome/robvis-visualization-tool ))

Response #1:

Thank you for helpful comment. As you advised, I re-evaluated the risk of bias of included clinical trials using RoB 2.0 (Figure 3) and add in the results section (2.3 Risk of Bias (RoB)).

Comment #2:

Because this is a systematic review, the most important step is to check the method section before reading the rest of the data. Because of that, please provide the full name of the abbreviation used in the method section or, alternatively, provide the list of used abbreviations at the beginning of the manuscript.

Response #2:

A list of used abbreviations is provided before the beginning of the manuscript.

Comment #3:

Please formulate the research question and provide it with the overall aim of the study at the end of the introduction.

Response #3:

At the end of the introduction, I added the research question and overall aim of this study.

Comment #4:

Because the authors present a lot of data, especially forest plots, I believe Table 1 could be moved into the supplementary file.

Response #4:

Thank you for good comment. I moved Table 1 to “Supplementary 1. Characteristics of included study”

Comment #5:

Table 1 – please define A and B in the Note below the table. Providing it only in the column description is insufficient. Moreover, because this table is huge, the first row should also be displayed on every page.

Response #5:

Added definitions for A and B in the Note below the table. Also, as advised, I added the first row at the beginning of each page table.

Comment #6:

Please provide content from Table 2 in the form of some figure/graph for better display.

Response #6:

I added “Figure 2. Frequency of usage of herbs in herbal medicine prescriptions (Top 10)” below the Table 2.

Comment #7:

2.3 RoB section needs to be rewritten to meet RoB 2.0 standard.

Response #7: After re-evaluated the risk of bias of included clinical trials using RoB 2.0, I rewrote it accordingly.

Comment #8:

Figure 2 – needs to be changed for RoB 2.0

Response #8: After re-evaluated the risk of bias of included clinical trials using RoB 2.0, I made a Figure 2 accordingly.

Comment #9:

Figure 3 – because I2 is below 50% in this situation, a fixed effect model with the Mantel-Haenszel method should be used. Other statistics are correct regarding using a Random model with inverse variance.

Response #9:

The studies were assumed to be a random sample of observable effect sizes because the clinical characteristics of the patients with depression across the studies were significantly heterogeneous. The data were pooled using a random-effects model regardless of heterogeneity, based on the I² statistic. However, the data were pooled using a fixed-effects model if very few studies were included in the meta-analysis, indicating that the estimate of the between-study variance lacked precision (4.5.1. Strategy for data synthesis in page 21). Nevertheless, if you consider using a random model is proper, we will follow your decision.  

Comment #10:

All of the RevMan results – in The RevMan you could change the name of the columns in the table section to be the same as in the plot section, eg. now you have Experimental/Control and Favours AD/Favours HM+AD – it should be the same.

Response #10:

I corrected the name of the columns in “Figure 5. Forest plot of the comparison between herbal medicine versus antidepressant assessing adverse effective rate” in “Supplementary 3. Forest plot for comparison of HAMD score and adverse effective rate.”

Comment #11:

Figures 6 and 7 – please use other software or omit this section completely because in RevMan when the random model is used, the funnel plot does not properly display the plot area. We are not able to see any funnel. 

Response #11:

Thank you for the comment. I performed a new funnel plot and Egger’s test using the Stata program.

Reviewer 2 Report

The manuscript ID: pharmaceuticals-2529722 entitled “Neuroendocrine Biomarkers of Herbal Medicine for Major De-pressive Disorder: A Systematic Review and Meta-Analysis” is systematic review autors Hye-Bin Seung et al. based on the systematic literature review and meta-analysis of herbal medicine treatment alone or with antidepressants  for MDD using neuroendocrine biomarkers.

My suggestion is to make minor changes

In the data summarized in Table 2, the names of that are traditionally used are listed. I think it is necessary to mention the Latin names for example Cnidii Rhizoma - Cnidium officinale Makino.

The authors themselves analyzed the problems limitations which are visible in this kind of meta-analysis.

In my opinion, the manuscript is well written, the data presented in this are interesting, and my suggestion is to accept it for publication in Pharmaceuticals.

Author Response

Comment #1:

In the data summarized in Table 2, the names of that are traditionally used are listed. I think it is necessary to mention the Latin names for example Cnidii Rhizoma - Cnidium officinale Makino.

Response #1:

Thank you for your insightful comment. According to your suggestion, I changed the traditional herbal names to Latin names.

"and one granule (44). Analysis of the frequency of single herbs according to HM composition showed that Bupleurum falcatum Linné and Glycyrrhiza uralensis Fischer were the most commonly used, followed by Paeonia lactiflora Pallas, Poria cocos Wolf, and Angelica gigas Nakai (Table 1)."

Table 1. Frequency of usage of herbs in herbal medicine prescriptions (more than once)

Frequency

Herb

21

Bupleurum falcatum Linné, Glycyrrhiza uralensis Fischer

20

Paeonia lactiflora Pallas

16

Poria cocos Wolf

10

Angelica gigas Nakai

9

Cyperus rotundus Linné

8

Atractylodes japonica Koidzumi, Citrus aurantium Linné, Citrus unshiu Markovich, Pinellia ternata Breitenbach

7

Curcuma aromatica Salisb, Polygala tenuifolia Willdenow, Zizyphus jujuba Mill

6

Cnidium officinale Makino

5

Scutellaria baicalensis Georgi 

4

Citrus aurantium Linné, Gardenia jasminoides Ellis, fossilia ossis mastodi

3

Albizia julibrissin Durazz, Aucklandia lappa Decne, Polygonum multiflorum Thumb, Cinnamomum cassia Presl, Ostrea gigas Thunberg, Lilium lancifolium Thunb, Hypericum perforatum, Paeonia suffruticosa Andrews, Panax ginseng C. A. Meyer, Rheum palmatum Linne, Zizyphus jujuba Miller var. inermis Rehder

2

Aconitum carmichaeli Debeaux, Acorus gramineus Solander, Astragalus membranaceus Bunge, Platycladus orientalis Franco, Codonopsis pilosula Nannfeldt, Mentha arvensis Linné var. piperascens Malinvaud ex Holmes, Salvia miltiorrhiza Bunge, Zingiber officinale Roscoe

Round 2

Reviewer 1 Report

Comments in the file

Author Response

Response to reviewer.

Comment #1.
"I am satisfied with the corrections made, with one exception. Please also change the descriptions of the column names in RevMan tables to be the same as on the forest plot (which I highlighted earlier in comment 10). You could define the column's name in RevMan. In the below figure, you should provide AD or HM (like on plot) instead of experimental and control (in table). It would help if you had the same description because it could confuse which one is experimental and which is the control."

Response #1.
Thank you for your kind comment and sorry for my uncarelessness.  The experimentaI group is HM and control group is AD. I have corrected all figure 4~7 including forest plots(figures and supplementary 3).